# Energy Productivity and Environmental Degradation in Germany: Evidence from Novel Fourier Approaches

Kwaku Addai [1] , Rahmi Deniz Ozbay [2], Rui Alexandre Castanho [3,4] , Sema Yilmaz Genc [5], Gualter Couto [6] and Dervis Kirikkaleli [7,*]

1   Department of Business Administration, Faculty of Economic and Administrative Sciences, European University of Lefke, Lefke 99770, Turkey
2   Department of Economics, Faculty of Business, İstanbul Commerce University, Istanbul 34445, Turkey
3   Faculty of Applied Sciences, WSB University, 41-300 Dąbrowa Górnicza, Poland
4   College of Business and Economics, University of Johannesburg, Auckland Park, P.O. Box 524, Johannesburg 2006, South Africa
5   Department of Economics, Faculty of Economics and Administration Sciences, Yildiz Technical University, Istanbul 34220, Turkey
6   School of Business and Economics and CEEAplA, University of Azores, 9500-321 Ponta Delgada, Portugal
7   Department of Banking and Finance, Faculty of Economics and Administrative Sciences, European University of Lefke, Mersin 10, Lefke 99010, Turkey
*   Correspondence: dkirikkaleli@eul.edu.tr

**Abstract:** The increased consumption of fossil fuels worldwide has resulted in unprecedented historic environmental degradation and global warming. According to the United Nations, this is both the defining crisis of our time and a race the world could win given the right policy attention. Researchers seek to find critical pathways to provide policy recommendations for reducing environmental degradation. This paper aims to investigate the effect of energy productivity on environmental degradation in Germany while controlling for economic growth, primary energy consumption, and globalization for the period between 1990Q1 and 2019Q4. The outcomes of the Fourier ARDL long-run estimates indicate that (i) both energy productivity and globalization have a negative effect on carbon emissions in Germany, and (ii) both economic growth and primary energy consumption have positive effects on carbon dioxide emissions. These outcomes provide significant policy insights to EU members with respect to reducing their reliance on Russian energy imports amidst the rising energy bills and ongoing geopolitical war with Ukraine while increasing investments to realize their energy turnaround policy objectives.

**Keywords:** Germany; Fourier ARDL; Fourier ADL cointegration; Fourier ADF unit root; energy productivity; environmental degradation; globalization; primary energy consumption; growth





## 1. Introduction

Despite the central role played by energy in growth and global economic prosperity, its biggest drawback is the contributions it makes to global warming and climate change [1]. Global warming has been found to have negative impacts on human society [2]. According to [3], excessive energy consumption threatens the global ecosystem. Several empirical studies have validated the claim that the energy sector is one of the primary determinants of global carbon dioxide emissions and accounts for approximately two-thirds of greenhouse gas emissions [4]. Among the factors found to be causing the rising levels of energy use are population growth, industrial productivity, economic growth, transportation, and infrastructural development [5]. It has been found that energy sector decarbonization is urgently needed in light of the increasing global issues of climate change and environmental degradation. Experts have warned that serious rethinking is needed with regard to energy use to ensure the global economy pursues economic growth that is sustainable [6]. For this purpose, the escalating dependency on an energy system dominated by fossil fuels is

ill-considered and needs rethinking. Factors of particular significance are greenhouse gas emissions and pollution, two negative externalities of fossil energy production and use.

Given this need for rethinking the human activities causing environmental degradation, global institutions and policymakers have intensified their efforts to find energy pathways that generate and assure sustainable use through investments in research and technology [7,8]. Efforts have equally been made to reduce emissions and switch to renewable energy sources, which are considered to be essential global actions toward addressing the challenges resulting from the growing energy demand. From solar, hydro, tidal, geothermal, and biomass energy use, energy productivity has been identified as the most significant factor responsible for an effective energy transformation. An approximate 3% increase in energy productivity investments was globally recorded in 2017 compared to 2016. As energy demand is expected to grow by 90% in emerging economies by 2035, several economies and international organizations are focusing their efforts on energy productivity as a tool to curb energy-related environmental pollution [8,9]. It continues to be debated in the literature whether the environmental effects of energy productivity could be validated and whether there are other variables that determine carbon emissions in Germany.

To resolve this debate, the economy of Germany presents an interesting and unique case for such an investigation into the effects of energy productivity on environmental degradation [10]. Given that Germany has a large and developed economy, ranked fourth in the world by nominal GDP, one would expect the findings of this study would produce suggestions for global policy responses on energy productivity toward carbon emissions reduction. In terms of trade, Germany's exports are reported to have risen by EUR 10.5 B (9.5%) between April 2021 and April 2022, from EUR 111 B to EUR 122 B, while imports increased by EUR 24.8B (25.5%) from EUR 97.1B to EUR 122B. By the end of 2018, energy intensity stood at 0.98 kWh. According to statistics, as of 2020, Germany was ranked seventh in global primary energy consumption, with 100% of the population having electricity access, adding value to modern life. By the end of 2020, the carbon intensity of energy produced was 0.19 kg/kWh. (source.tradingeconomics.com (accessed on 7 of October, 2022).). As of 2021, the primary energy consumption of Germany increased to 12,193 petajoules, while approximately 75% of consumption was from fossil sources, 6.2% was sourced from nuclear energy, and 16.1% was sourced from renewables. Approximately 24.39% of German's energy mix comes from low-carbon sources (September 2022). Of these, 19.45% are renewables, while 4.93% are generated from nuclear power. Germany's energy consumption as a percentage of GDP has increased from 47.15 (2021Q1) to 51.97% (2022Q2). Globally, Germany ranks 14th in energy imports, accounting for 61.4% of its total energy consumption. The economy ranks 16th in global economic freedom. With uncertainty still surrounding the outcome of the Russia–Ukraine war, and to continue its commitment to the Paris agreement, Germany has intensified its energy transition pace with the implementation of new policies. The key concept behind the energy policy of Germany is "energy turnaround or transformation". The energy policy actions include a nuclear power phase-out by 2022, energy productivity, and a gradual replacement of fossil-based fuels with renewable-based energy. Germany's energy policy ensures investment in coal electricity production and relies on electricity imports to phase out nuclear power. Germany's "Energy Efficiency Strategy 2050" was adopted in December 2019, which outlines new energy efficiency targets for 2030 (to reduce primary energy by 30% by 2030 compared to 2008). On the 10th of June, 2020, the "National Hydrogen Strategy" of Germany was also adopted. Figure 1 shows the country's current energy mix, while Figure 2 is the Energy Consumption Fact Sheet of Germany.

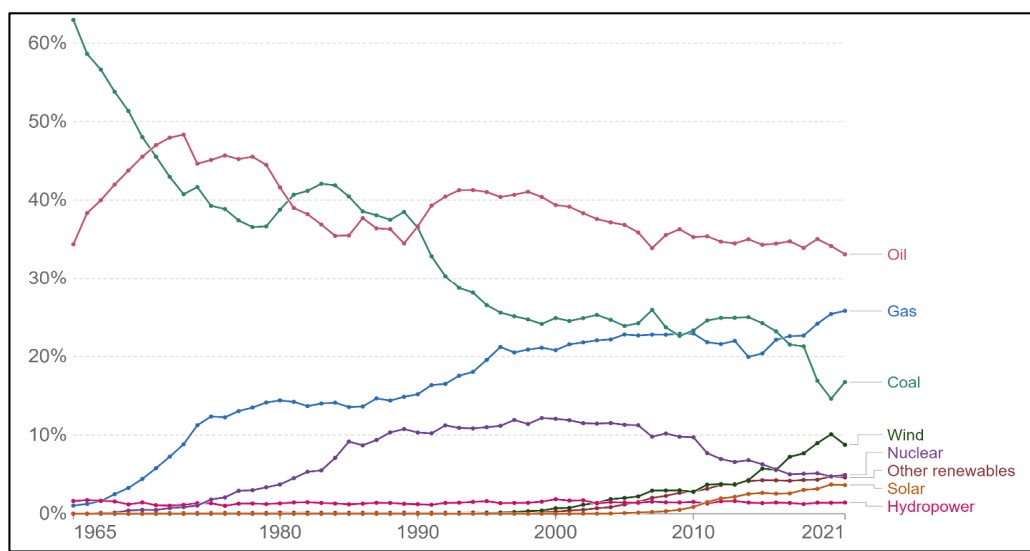

**Figure 1.** Germany's energy mix. Source: Our World in Data, 2022.

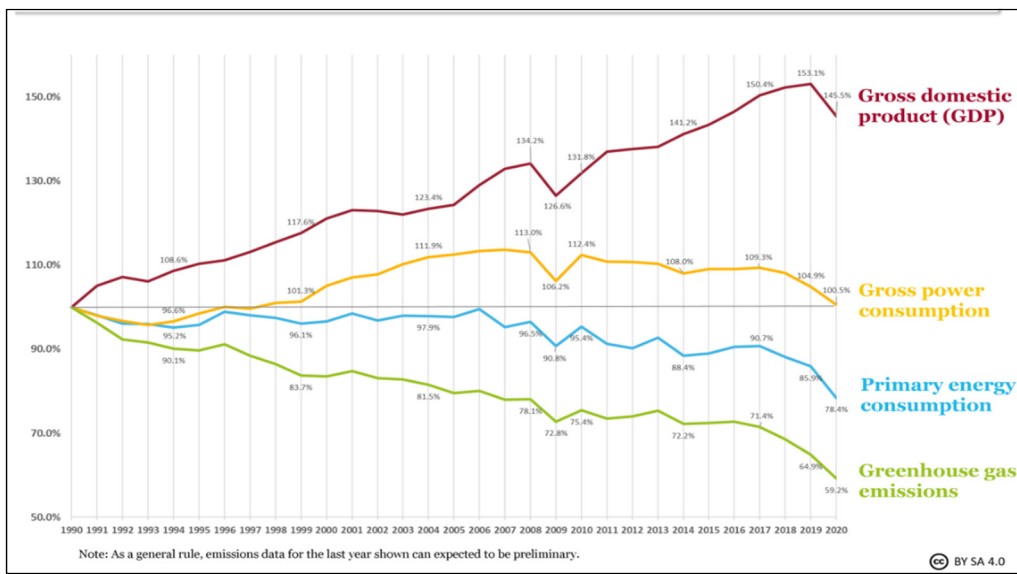

**Figure 2.** Growth, power, energy consumption, greemhouse gas emissions of Germany. Source: Our World in Data [11].

Given the rising energy prices as a result of the Russia–Ukraine war, it has been debated in the literature whether energy productivity can reduce carbon emissions. For example, in their recent study, ref. [12] found that energy prices have a positive effect on energy productivity in the petrochemical industry of Iran. A study by [13] found that in the UK and Germany, prices of energy exert a negative effect on carbon emissions since their impact on employment levels and production is severe. This finding was corroborated by [14], who found that the substitution effects of rising energy pricing were high, thus reducing economic output. With respect to the rising energy prices in Germany, the IMF (report in July 2022) forecasts that a significant reduction in economic activity, which will reduce GDP (relative to baseline levels) by approximately 1.5%, 2.7%, and 0.4% in 2022, 2023, and 2024, respectively, as well as a mean reduction in carbon emissions. To respond to the rising energy prices, the German government implemented new policies and programs in July 2022, which included steps to design new gas auction models to encourage industrial gas savings. Questions still linger whether the emergency policy actions on energy use

across the world due to the Russia–Ukraine conflict are attempts to diversify energy use or completely decarbonize.

Based on this information, this paper aims to identify the effect of energy productivity on environmental destruction in Germany while controlling for economic growth, primary energy consumption, and globalization in Germany from 1990Q1 to 2019Q4. Due to the empirical weaknesses of existing scholarly assessments, we employ dynamic and novel Fourier-based ARDL approaches. Fourier-based ARDL methods have several unique qualities. First, the integration order of regressors is not required. Additionally, the test can detect several seamless structural changes in time-series data. Fourier-based ARDL approaches can overcome two limitations identified in the empirical literature as (i) the approaches assume an unknown number of structural changes; (ii) the approaches can detect abrupt structural changes by using dummy variables [15]. Taking into account the rising energy prices, the outcomes of the study could offer significant policy insights to EU members in taking drastic policy actions when faced with the weaponization of energy by Russia. The study is further structured as follows: Section 2 is the literature review, Section 3 presents a description of the research methodology, while Sections 4 and 5 include the findings and conclusions, respectively.

## 2. Reviews of Published Works of Literature

Globally, the rate of economic growth is rising at levels detrimental to the environment due to the increasing levels of $CO_2$ emissions caused by the excessive use of fossil fuels for industrial production. According to several scholars, $CO_2$ emissions constitute a large proportion of greenhouse gas emissions; therefore, using $CO_2$ emissions as a measure of environmental quality is appropriate [16]. Scientific evidence suggests that increased concentrations of carbon dioxide emissions have serious consequences for air, water, forest, and land resources. Countries across the world are using various eco-innovation approaches to deal with environmental degradation. The term "eco-innovation" refers to the creation of procedures and solutions that support environmentally-friendly manufacturing and consumption through the commercial application of science and technology. This encompasses a broad spectrum of related ideas, theories, and skills from sustainable technologies. Through eco-innovation initiatives, corporations have been able to increase their profitability and fuel efficiency [17]. Eco-innovation enables economies to employ eco-friendly and cost-effective technologies for sustainable consumption and production practices. Cleaner production methods have been found to lessen the detrimental effect of industrial and economic progress on the environment [18]. Several empirical studies have identified the significance of eco-innovation methods in reducing $CO_2$ emissions. Additionally, environmental economists have found that eco-innovation approaches can enhance corporate performance, reduce energy intensity, and improve environmental quality.

An indication of the economic benefit for consuming a unit of energy is called energy productivity. It is determined by dividing the total energy utilized by economic output. Energy productivity differs from energy intensity, which represents the energy used for production per unit of GDP. Energy productivity helps reduce energy costs and improves environmental quality [19]. Additionally, increased energy productivity results in higher economic production with less energy use. Energy productivity controls $CO_2$ emissions in three ways. First, less energy is needed for the same economic activity when energy productivity improves. Energy productivity also results in a decrease in energy expenses. Third, because oil imports are reduced because of improved energy productivity, carbon dioxide emissions are also reduced. To achieve sustainable development, several economies have prioritized energy productivity in their energy policies. Today, countries can estimate energy consumption levels using energy productivity tools. Policymakers seeking to invest in energy technology consider the benefits of energy productivity. Table 1 shows a list of some relevant studies in the field.

**Table 1.** Literature Review Summary.

| Authors | Country/Context and Period | Technique(s) Used | Findings |
|---|---|---|---|
| [20] | Mexico 1970–2016 | ARDL, Granger causality | (+)GDPg and $CO_2$E |
| [21] | Thailand 1970–2016 | ARDL and Granger causality | (+) GDPg causes (+) $CO_2$E |
| [22] | India 1970–2019 | Wavelet coherence | (+) GDPg and $CO_2$E |
| [23] | China 1965–2019 | PWC, MWC, WC | (+) GDPg and $CO_2$E |
| [24] | APEC Nations 1960–2016 | CUP, panel causality | GLB and $CO_2$E |
| [25] | South Africa 1980–2018 | ARDL, frequency domain | (+) GLB and $CO_2$E |
| [26] | 155 emerging and advanced countries 1991–2018 | GMM | (+) GLB and $CO_2$E |
| [27] | Brazil and China 1971–2016 | Fourier ARDL | GLB and $CO_2$ (+) |
| [28] | France 1960–2003 | ARDL | PEC and $CO_2$E |
| [29] | UAE 1990–2008 | GRAPE | PEC and $CO_2$E (+) |
| [30] | 23 emerging countries 1993–2016 | IV GMM | U-shaped relationships |
| [31] | G-7 1980–2014 | GMM | PEC raises $CO_2$E emissions; financial globalization reduces $CO_2$E |

Notes: GLB is globalization, $CO_2$E is carbon dioxide missions; PEC is primary energy consumption; GDPg is GDP growth.

To understand the range of environmental degradation in Germany from an empirical perspective, the following research questions are explored in this study: (i) Does energy productivity play a role in explaining the reduction in $CO_2$ emissions in Germany?; (ii) Is there any environmental effect on economic growth, primary energy consumption, and globalization in Germany? Answers to these questions are crucial for Germany and the EU, as (i) it may support the process of forming policies to reduce carbon emissions, and (ii) the findings could help provide significant inputs for future academic research on environmental degradation. Accordingly, the following hypotheses are proposed:

**Hypothesis 1 (H1).** *Economic growth has positive effects on carbon emissions in Germany.*

Due to the uncontrolled rates of economic growth in recent decades, the world has seen unprecedented increases in global greenhouse gas emissions. Several empirical studies have confirmed the harmful externalities of economic expansion on the environment. Less developed economies lack the technological capabilities to balance environmental quality promotion and economic growth. Industrial development in developed economies has accelerated growth and caused an ever-increasing demand for fossil fuels, leading to an increase in environmental pollution. In their study on the asymmetric effect of clean and unclean energy on environmental quality in less developed countries, ref. [32] found that unclean energy utilization exerts an inverted U-shaped effect on transportation-based environmental quality, but poses a U-shaped effect on industry-based environmental quality.

For example, in their study on energy use, industrial development and environmental quality, found that fossil fuel use deteriorates the quality of the environment in China. An empirical study that assessed the relationship between economic growth and carbon dioxide emissions in South Africa by [33] indicated an increase in carbon emissions during a period in which the economy saw rising growth rates. Accordingly, we assume that a steady increase in economic growth significantly increases $CO_2$ emissions in Germany, i.e., $\vartheta_1 = \frac{\vartheta LCO_2E}{\vartheta LGDP} > 0$.

**Hypothesis 2 (H2).** *Primary energy consumption has positive effects on carbon emissions in Germany.*

Ensuring energy security is channeled through the promotion of cleaner energy resources. Approximately 64% of the total primary energy supplies of Germany are imported, while nuclear energy is the second-lowest component of primary energy, with oil being first, followed by gas and coal. However, the shares of renewable and nuclear energy in the primary energy mix have slightly increased in the last couple of decades in Germany. Ref. [7] studied the role of trade in the nexus between energy use, carbon emissions, and economic growth in South Africa. The outcomes indicated that per capita use of primary energy had a positive effect on carbon emissions. Ref. [34] investigated the effect of primary energy utilization on $CO_2$ emissions and found it to be positive in sixty economies from 1965 to 2016. Based on these, we assume that a steady increase in primary energy consumption reduces $CO_2$ emissions in Germany, i.e., $\vartheta_2 = \frac{\vartheta LCO_2E}{\vartheta LPEC} > 0$.

**Hypothesis 3 (H3).** *Globalization has negative effects on carbon emissions in Germany.*

Globally, scientists have found that increasing economic expansion, urbanization, and globalization are some of the key variables affecting environmental quality. Additionally, industrial processes affect energy consumption. Further, if the investments are directed toward environmentally-friendly sectors and technologies, there will be no harm to the environment [35,36]. By controlling for capitalization, financial development, and financial globalization, ref. [31] examined the impacts of biomass energy consumption on $CO_2$ emissions. The outcomes indicated that financial globalization improved environmental quality. By using a cross-correlation framework, ref. [37] investigated a globalization-driven carbon dioxide emissions hypothesis in 87 economies. Although the outcomes validated the inverted U-shaped EKC hypothesis, they also indicated a U-shaped linkage between globalization and environmental degradation for seven of the investigated economies. Theoretically, economic globalization affects $CO_2$ emissions and economic activity through different channels. Trade engagements and investment activities result in increased energy use for production, which culminates in increased carbon emissions. Environmental degradation results directly from the technology employed for the generation of economic output [37]. If energy-intensive and inefficient technologies are used for production, economic growth will lead to environmental quality destruction [9]. Additionally, globalization has been found to create economic integration, which facilitates technology innovations and investment capital. This enables the growth of collaborating economies with high environmental costs. This confirms globalization's scale, composite, and technique effects on carbon emissions. Based on these reviews, the study assumes that a steady increase in financial globalization reduces $CO_2$ emissions in Germany, i.e., $\vartheta_1 = \frac{\vartheta LCO_2E}{\vartheta LGLO_{it}} < 0$.

**Hypothesis 4 (H4).** *Energy productivity has negative effects on carbon emissions in Germany.*

The increasing need for global economies to promote energy productivity through technology has taken center stage in international policy discussions over the last decade due to the increasing threats posed by the irresponsible use of energy. Several developed economies, including Germany and the UK, were among the initial crop of economies to

introduce energy productivity policies and launch green patent applications beginning from 2009. Energy productivity investments provide strong economic viability for cost reduction, increased innovation into green technologies, and cause reduction in energy intensity [38]. Nevertheless, critics of energy prodictivity argue, at the macro-level, energy productivity initiatives do not present 'win–win' policy outcomes, because of the "income-responsive" energy rebound effect [39,40]. Historically, ref. [41] demonstrated that improving energy productivity with a Cobb–Douglas function within a neoclassical growth framework does not result in energy savings. Increasing energy consumption is linked to the rate of increase in output. Therefore energy productivity which ultimately leads to a decrease in energy demand (consumption) will cause output to fall. In response, proponents of the energy productivity debate have recognized that Sounder's theoretical argument is limited because the income effect from energy productivity improvements is insignificant compared to gains from technology processes, including learning-by-doing, R&D investments, and technical progress [42,43]. In their investigations in Saudi Arabia on the effects of energy productivity measures between 2000 and 2004, ref. [42] found that rudimentary energy retrofitting through energy efficiency and productvity policies in residential housing generated significant environmental and economic gains. Similarly, a study by [13] indicated that Germany's tax policies on energy use in exerted negative effect on carbon emissions in 2009. Based on these arguments, the study assumes that a steady investment in energy productivity significantly decreases $CO_2$ emissions in Germany, i.e., $\vartheta_4 = \frac{\vartheta LCO_2E}{\vartheta LEPR} < 0$.

Based on the literature review, the environmental effects of energy productivity, economic growth, primary energy consumption, and globalization need further investigation for the case of Germany. These four variables are major validated factors used by several scholars in the literature to account for environmental degradation. In particular, the academic arguments on the relationship between energy productivity and environmental degradation remain largely unresolved. Additionally, it appears that the environmental effects of energy productivity remain predominantly unexplored.A specific area that has not been investigated is the relationship between energy productivity and environmental degradation by using novel Fourier-based ARDL estimators while keeping the environmental implications of economic growth, globalization, and primary energy consumption under control. By investigating this relationship in the case of Germany, the study will likely bring clarity and closure to the ongoing debate in the literature about the environmental gains of energy productivity. Over the years, studies have found how variables, such as energy, environment, and economic growth, are interconnected with carbon emissions. However, few studies have focused on how energy productivity determines carbon emissions. Therefore, this study seeks to contribute to the literature on the two variables.

### 3. Data and Methodology

Data used and sources:

This study assesses the effect of energy productivity on environmental degradation in Germany between 1990Q1 and 2019Q4 while controlling for economic growth, primary energy consumption, and globalization. Data were collected on (i) economic growth (World Bank), which is an indicator of economic and industrial activity, determining the level of $CO_2$ emissions, sourced from the World Development Indicators; (ii) primary energy consumption, which was sourced from UNFCC; (iii) globalization [37], which was sourced from the KOF globalization index, is the sum of economic, social, and political globalization; (iv) GDP per unit of total primary energy supply (TPES, USD/toe) to serve as a proxy for energy productivity [4], which was sourced from Eurostat; (v) $CO_2$ emissions (as a proxy for environmental degradation), which was sourced from the UNFCC database. $CO_2$ emissions [10] are regarded as the largest contributor to greenhouse gas emissions; therefore, several scholars assert that using $CO_2$ emissions as an environmental quality indicator is appropriate [44,45]. Economic growth, primary energy consumption, and globalization were selected as the explanatory variables based on theories and empirical evidence show-

ing their participation in generating and intensifying carbon dioxide emissions [46,47]. To avoid scaling, all variables were logged. The analyisi flowchart is reported in Figure 3.

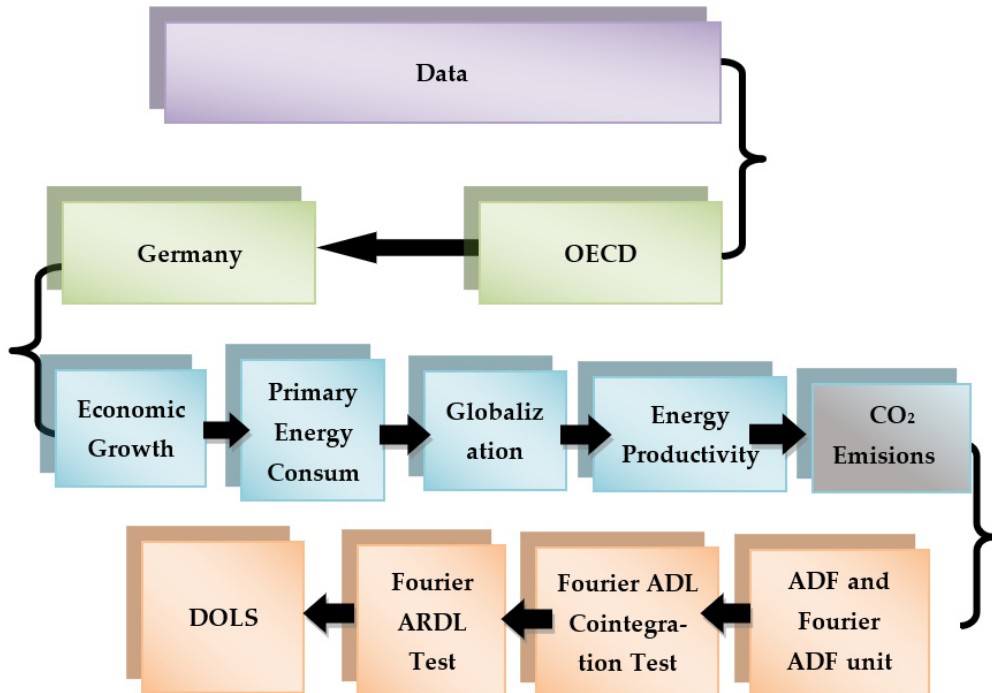

**Figure 3.** Analysis flowchart.

### 3.1. Theoretical Foundation

As global concerns about climate change and environmental degradation grow, new pathways must be found for a successful energy transformation. Energy productivity is key to saving energy and reducing carbon emissions. Theoretically, efficient production and the productive utilization of energy promote environmental quality [43]. According to them, energy productivity results from energy savings from efficient efficient use, which could have multiple implications to the national grid, including the demand side and responsive management. Recent scholars have used energy productivity to assess energy efficiency gains of an economy. However, ref. [43] claim that energy productivity is applicable to determining not only financial and social factors, but also the environmental value gained from energy consumption. Similarly, economic globalization facilitates investments in efficient energy technologies that propel sustainable growth, which substantially generates reduction in environmental degradation [37]. Studies have also recorded increasing carbon emissions in primary energy consumption [48]. To determine the relationship between the dependent and the independent variables sourced, the study adopts the Green Solow model, which establishes a theoretical basis for predicting the emissions, emissions intensities, and pollution abatement in an economy. The framework can predict an EKC relationship between carbon emissions, income per capita, and other variables [49]. There are two approaches by which productivity can be analyzed (i.e., traditional and frontier approaches). The standard neoclassical production models initially used by Solow were purely focused on economic variables (i.e., capital and labor) and ignored natural resources. The empirical model for this study is based on the stochastic frontier approach, which identifies technological frontier effects where "fuel consumption" is required for economic activity and in the process pollutes the environment. This model had been used to successfully assess sustainability efficiency of the transport industry in China in relation to its $CO_2$ emissions. Additionally, the stochastic frontier model is extendable to cater for effects of covariates, around the cost function. It assumes each observation could be added to a vector of covariates [8]. It case the frontier model could be employed to assess the linkages

between energy productivity, economic globalization, primary energy consumption, and pollution. Therefore, the empirical model for this study is:

$$LCO_{2it} = \beta_1 LGDP + \beta_2 LGLO + \beta_3 LEPR_t + \beta_4 LPEC_t + \varepsilon_{it} \tag{1}$$

where $LCO_2$, $LGDP$, $LGLO$, $LEPR$, and $LPEC$ stand for production-based $CO_2$ emissions, economic growth, globalization, energy productivity, and primary energy consumption, respectively.

### 3.2. Fourier ADF and ADF Unit Root Tests

After reviewing the descriptive statistics, we check the integration order of the time-series variables using Fourier ADF and ADF unit root tests with breakpoints. Historically, several scholars have used different approaches for computing unit roots for time series. In econometric analysis, it has been found that variables are integrated at varying degrees, meaning that models are not testable for cointegration using traditional cointegration approaches. Therefore, we adopted the Fourier ADF and ADF unit root tests, which produce more reliable results compared to traditional unit root tests and allow cointegration analysis regardless of the order of integration (i.e., I(0)/I(1) or mixed orders). To estimate the unit roots for both ADF with breaks and FADF, the null hypothesis of the unit root is formulated as:

$$x_{t=} \mu + \rho x_{t-1} + e_t \tag{2}$$

where $x_t$ refers to interest variables, $\mu$ is the constant term, and $e_t$ is the error term. By taking the first differencing, the equation becomes $\Delta x_{t=} \mu + e_t$, where $\Delta = (1 - B)$, $\rho$ is the slope parameter for the lagged dependent variables, which becomes 1 if there is a unit root. The alternative unit roots for ADF with breaks and FADF are shown in Equations (3) and (4), respectively:

$$x_t = \mu + \beta t + \gamma_1 \sin(2\pi kt/N) + \gamma_2 \cos(2\pi kt/N) + e_t \tag{3}$$

$$x_t = \mu + \beta_t \, \delta DU_t + \theta D(T_B)_t + \varepsilon_t \tag{4}$$

where $\beta$ is the slope parameter for the trend, k is the Fourier frequency, $\gamma$ is the slope parameter in the Fourier function, t is the trend, N is the number of observations, $\pi = 3.1416$; $\delta$ is the slope parameter for the structural break dummy ($DU_t = 1$, if $t > T_B$ and $DU_t = 0$, if otherwise), $T_B$ is the breakpoint when a structural break occurs, $\theta$ is the slope parameter for the one-time break dummy, ($D(T_B)_t = 1$ if $t = T_B$ and $D(T_B)_t = 0$ if otherwise) [6].

Having re-written these models in error correction form and by including the augmentation component, we estimate these equations for ADF with breaks and FADF in Equations (5) and (6), respectively, as:

$$\Delta x_t = \mu + \beta_t + \gamma_1 \sin(2\pi kt/N) + \gamma_2 \cos(2\pi kt/N) + (\rho - 1)x_{t-1} + + \sum_{i=1}^{P} ci\Delta x_{t-I} + \varepsilon t \tag{5}$$

$$\Delta xt = \mu + \beta t + \delta DUt + \theta D(T_B)t + (\rho - 1)x_{t-1} + \sum_{i=1}^{P} ci\Delta x_{t-I} + \varepsilon_t \tag{6}$$

where c is the slope parameter of augmented components, and p is the lag length for augmentation determined by minimum information criteria values. More information on the selection of the optimal *k* Fourier frequency, structural break date $T_B$, and break fraction $\lambda$ can be found in [50] which proposed the model fitness test (F test), which has restricted and unrestricted models. The ADF regression is a restricted model to (i) ADF with a structural break if there is no structural break; (ii) FADF if the parameters for Fourier nonlinearity are insignificant in the FADF estimates; (iii) FADF with a structural break if both nonlinearity and structural breaks are absent.

The method assumes that unknown nonlinearities of time series, including structural changes, could be accurately detected using the low-frequency components of a Fourier approximation. This is because breaks and structural changes shift spectral density functions towards zero frequencies.

### 3.3. Fourier ADL Cointegration and Fourier ARDL Tests

The autoregressive distributed lag (ARDL) cointegration method has been used by various researchers for decades. However, we tested the existence of cointegration by using the Fourier ADL cointegration test, which considers unknown structural breaks, time, and structure. The results provided by this method are more effective than those produced by VECM analysis.

The Fourier-based ARDL method provides a more robust long-run estimation outcome than the traditional ARDL approach. Additionally, Fourier functions can identify structural changes, although, for the Fourier-based ARDL approach, a further structural changes test is not required. The Fourier function created by [51] considers structural changes in the model, as seen in Equation (2).

$$d(t) = \sum_{k=1}^{n} ak \sin\left(\frac{2\pi kt}{T}\right) + \sum_{k=1}^{n} bk \cos\left(\frac{2\pi kt}{T}\right) \tag{7}$$

where 'n' indicates the number of frequencies, $\pi = 3.14$, 'k' is the number of special frequencies selected, 't'is the trend, and 'T' is the sample size. A single frequency value suggested is used in Equation (8).

$$d(t) = \gamma_1 \sin\left(\frac{2\pi kt}{T}\right) + \gamma_2 \cos\left(\frac{2\pi kt}{T}\right) \tag{8}$$

The FARDL model for this study is shown in Equation (9).

$$\Delta LCO_{2t} = \beta_0 + \gamma_1 \sin\left(\frac{2\pi kt}{T}\right) + \gamma_2 \cos\left(\frac{2\pi kt}{T}\right) + \beta_1 LCO_{2t-1} + \beta_2 LGDP_{t-1} + \beta_3 LGLO_{t-1} + \beta_4 LEPR_{t-1} + \beta_4 PEC_{t-1} +$$
$$\sum_{i-1}^{\rho-1} \varphi_i' \Delta LCO_{2t-i} + \sum_{i-1}^{\rho-1} \delta_i' \Delta LGDP_{t-i} + \sum_{i-1}^{\rho-1} \varnothing_i' \Delta LGLO_{t-i} + \sum_{i-1}^{\rho-1} \vartheta_i' \Delta LEPR_{t-i} + \sum_{i=1}^{\rho-1} \vartheta_i' \Delta PEC_{t-i} + e_t \tag{9}$$

In addition, we used DOLS approaches to support the outcomes of the Fourier ARDL test.

### 4. Empirical Findings

This paper aims to capture the effect of energy productivity on environmental degradation in Germany while controlling for economic growth, primary energy consumption, and globalization in Germany from 1990Q1 to 2019Q4. The descriptive statistics of the times series varables are reported in Table 2. Besides having some level of environmental impact on the air, ground, and water, all forms of power generation emit some greenhouse gas emissions. The policy implications of efficient power production are significantly pervasive for Germany, as the findings of this study could provide valuable insights.

**Table 2.** Descriptive statistics.

| Description | Production-Based CO$_2$ Emissions | GDP (Constant 2015 USD) | Globalization Index | Energy Productivity | Primary Energy Consumption |
|---|---|---|---|---|---|
| Variables | LCO2 | LGDP | LGLO | LEPR | LPEC |
| Mean | 5.908057 | 12.46770 | 1.919907 | 4.013947 | 3.590470 |
| Median | 5.912115 | 12.46206 | 1.928233 | 3.994436 | 3.596066 |
| Maximum | 5.982226 | 12.55802 | 1.948486 | 4.158645 | 3.625561 |
| Minimum | 5.802051 | 12.35858 | 1.857351 | 3.873457 | 3.554276 |
| Std. Dev. | 0.037966 | 0.051486 | 0.024827 | 0.069455 | 0.016674 |
| Skewness | −0.217746 | −0.029198 | −0.657973 | 0.265238 | −0.375014 |
| Kurtosis | 2.593511 | 2.029170 | 2.065853 | 2.084236 | 2.132716 |
| Jarque–Bera | 708.9668 | 1496.124 | 230.3889 | 481.6736 | 430.8564 |
| Probability | 0.171530 | 0.315443 | 0.073351 | 0.574053 | 0.033084 |

Our next step is to check for unit roots in the series using the Fourier ADF unit root and ADF unit root with break tests. However, before employing the Fourier ADF unit root estimation, it is appropriate to check the statistical significance of the Fourier function. If the F-STAT of the time-series variables is found to be statistically significant, we can test the Fourier ADF unit root; otherwise, we choose the standard ADF unit root test with a breakpoint to test the integration of the order of the time-series variables.

The outcomes of the unit root test suggest that the time-series variables, except LEPR and LPEC, were not stationary at level, but LCO2, LGDP, and LGLO variables became integrated at order I(1) with several breakpoints as reported in Table 3. Based on the Fourier ADF unit root outcomes, LEPR is stationary at a 5% significance level, indicating that the variable is integrated at I(0). The outcomes of the unit root tests reveal that the time-series variables have a mixed order of integration.

**Table 3.** Fourier ADF and ADF Unit Root tests.

| Variable | F-STAT | FADF | ADF Break Trend |
|---|---|---|---|
| LCO2 | 0.488185 | | −3.758 (2019Q1) |
| LGDP | 1.836510 | | −4.678 * (2008Q1) |
| LGLO | 4.089875 | | −2.822 (1996Q1) |
| LPEC | 3.037139 | | −4.596 ** (2008Q1) |
| LEPR | 6.423204 *** | −3.726346 ** | |
| DLCO2 | | | −6.597 *** (1992Q1) |
| DLGDP | | | −7.193 *** (2009Q1) |
| DLGLO | | | −6.287 (1991Q1) |
| DLPEC | | | |
| LEPR | | | |

Note: *, **, *** indicate level of statistical significance at 10%, 5%, and 1%, respectively.

To estimate the long-run coefficients, we first conduct diagnostic tests on our empirical ARDL model to ensure that it is free from serial autocorrelation and heteroscedasticity, and the CUSUM graph indicates substantial stability. Subsequently, it is necessary to check the cointegration properties of the time-series variables using the Fourier ADL cointegration test. Traditionally, ARDL bounds testing refers to the use of the cointegration estimator to check the long-run relationship among variables regardless of their integration order (i.e., I(0) or I(1)). Second, the estimator can help derive the unrestricted error correction model (UECM) through very simple linear processes. We test the long-run linkage among the time-series variables by using the Fourier ADL cointegration test, which considers unknown structural breaks, time, and structure. The results of the ADL cointegration test (See Table 4) provide more robust results than those provided by the VECM approach.

**Table 4.** Fourier ADL Cointegration Analysis.

| Model | Test Statistics | Frequency | Min AIC |
|---|---|---|---|
| LCO2 = f(LEPR, LGDP, LGLO, LPEC) | −8.498597 *** | 1 | −8.662762 |

Note: *** denote 1% significance level. The decisions are based on Banerjee et al.'s critical values (2017).

As the order of integration of the variables in the times series is determined to be mixed, this allows the present study to implement Fourier ARDL-based models. As a next step, we check the cointegration relationship between the selected variables using the Fourier ADL cointegration test to capture the effect of energy productivity on environmental degradation in Germany. The outcomes reveal there is a long-run linkage in the estimated model. These outcomes support the findings of the recent study of [15].

The outcomes of the Fourier ARDL long-run estimates indicate that the coefficients of LEPR and LGLO are negative, implying that a one unit increase in both LEPR and

LGLO has negative effects on $CO_2$ emissions, as seen in Table 5. First, Germany has committed to an ambitious program to phase out nuclear and coal-sourced energy by 2050. To fulfil this commitment, huge budgetary allocations have been made to energy technology and productivity projects; therefore, it is rational to observe that the effect of energy productivity on environmental degradation in Germany is negative. These findings support the outcomes of [52]. Second, the outcome shows that LGLO has a negative effect on $CO_2$ emissions in Germany. This is because, for many years, Germany's economic transformation has promoted win–win economic globalization policies and attracted huge investments in sustainable sectors of the economy. The resultant effect of these policy is a reduction in the rate of carbon emissions. These findings support [53], who found that globalization through trade openness and energy productivity has negative effects on carbon emissions in the MENA region. Nevertheless, critics claim that the government of Germany could do better, especially in the transport sector. Similarly, ref. [54] have found that Volkswagen AG, the leading automobile manufacturer, has been caught in a diesel emission scandal in several countries, namely the US, Germany, and China, due to globalization. These findings on Volkswagen AG confirm earlier work by [5].

**Table 5.** Fourier ARDL Long-Run Form.

| Variable | Coefficient | Std. Error | t-Statistic | Prob. |
|----------|-------------|------------|-------------|-------|
| LGDP | 0.646570 | 0.384125 | 1.683230 | 0.0960 *** |
| LEPR | −1.269875 | 0.546158 | −2.325104 | 0.0225 ** |
| LGLO | −1.030950 | 0.353461 | −2.916732 | 0.0045 * |
| LPEC | 1.927779 | 0.495773 | 3.888431 | 0.0002 * |
| C | −1.464152 | 0.415509 | −3.523756 | 0.0007 * |
| CointEq(−1) * | 0.103567 | 0.015676 | 6.606778 | 0.0000 * |

Note: *, **, *** indicate level of statistical significance at 1%, 5%, and 10%, respectively.

The findings also show that both LGDP and LPEC have positive coefficients for $CO_2$ emissions, which implies that any unit upward adjustment in both LGDP and PEC causes a corresponding increase in $CO_2$ emissions. The finding for LGDP invalidates the EKC hypothesis, which states that the long-run effects of economic growth on $CO_2$ emissions fall. The nexus between economic growth and indicators of environmental indicated an inverted U-shape linkage between real per capita income and environmental pollution. They explained the relationship to mean that as economic growth increases, $CO_2$ emissions initially increase and thereafter start to decrease after reaching a certain turning point. It is, therefore, assumed that developed economies which have crossed this turning point, such as Germany, will see a fall in $CO_2$ emissions and not a rise [55]. This finding supports recent outcomes reported by [51] in their study of the effect of economic complexity on the ecological footprint in China using the EKC hypothesis. Similarly, the findings on the effect of LPEC showing positive coefficients for $CO_2$ emissions for Germany may largely be the result of the significant industrial dependence on coal and nuclear energy sources for electricity generation. This outcome of primary energy consumption in Germany also corroborates similar findings by [56], who found that the higher a country's national income, the higher the positive impact of primary energy consumption on $CO_2$ emissions. This means that primary energy consumption should be sacrificed to see a reduction in $CO_2$ emissions. Similarly, ref. [10] found that in the MENA region, increased $CO_2$ emission has resulted from the rising consumption of primary energy, which means a transition to green and low-carbon energy for this region is imperative. Therefore, Germany's continuous utilization of coal for industrial applications is concerning in terms of its effects on carbon emissions, and the economy must accelerate its plans to eliminate coal consumption. The outcomes in Table 6 show no heteroskedasticity, since the null hypothesis is not rejected,

while in Table 7, no serial correlation is observed. In addition, Cusum and Cusumsq figures are reported in Figures 4 and 5.

**Table 6.** Heteroskedasticity Test: Breusch–Pagan–Godfrey.

| F-statistic | 1.438756 | Prob. F (2, 85) | 0.1087 |
|---|---|---|---|

**Table 7.** Breusch–Godfrey Serial Correlation LM Test.

| F-statistic | 1.005954 | Prob. F (2, 83) | 0.3701 |
|---|---|---|---|

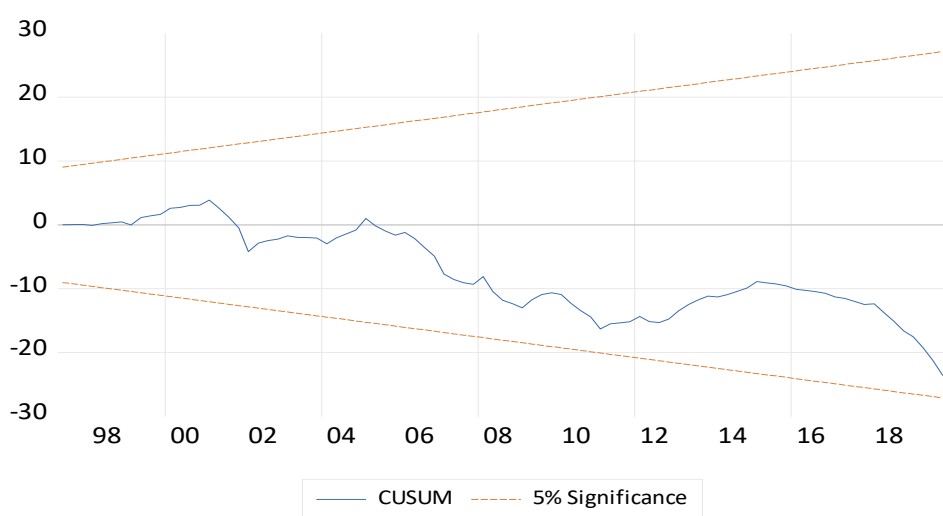

**Figure 4.** CUSUM.

**Figure 5.** CUSUMSQ.

The CUSUM (Figure 4) and CUSUMSQ (Figure 5) approaches employed in the Fourier ARDL model indicate that the coefficient estimates on the model are stable.

Following that, the robust least-square approaches based on DOLS are used to confirm the model outcomes (see Table 8). The outcomes provide an additional means of detecting symmetric long-run relationships and indicate that in the long run, while energy productivity and globalization reduce environmental degradation in Germany, energy consumption and economic growth increase $CO_2$ emissions. It is worth mentioning that the outcomes of the DOLS results are in line with the Fourier ARDL outcomes. According to the DOLS

estimates of model robustness (Table 8), the independent variables (i.e., LGDP, LEPR, LGLO, and LPEC) jointly explain 98.82% of $CO_2$ emissions in Germany. The outcomes are in line with the previous research by [57].

**Table 8.** Robustness Test.

| | DOLS | | | |
|---|---|---|---|---|
| Variable | Coefficient | Std. Error | t-Statistic | Prob. |
| LGDP | 0.411485 | 0.400428 | 1.027612 | 0.3069 |
| LEPR | −0.559247 ** | 0.264841 | −2.111633 | 0.0375 |
| LGLO | −0.660615 ** | 0.253006 | −2.611070 | 0.0106 |
| LPEC | 0.292323 | 0.267994 | 1.090781 | 0.2783 |
| C | 3.241625 | 2.648990 | 1.223721 | 0.2243 |
| R-squared | 0.988256 | Mean dependent var | | 5.907899 |
| Adjusted R-squared | 0.985125 | SD dependent var | | 0.034455 |
| SE of regression | 0.004202 | Sum squared resid | | 0.001589 |
| Long-run variance | $4.98 \times 10^{-5}$ | | | |

Note: ** indicate levels of statistical significance at 5%.

## 5. Discussions

This study has empirically investigated the effects of energy productivity on environmental degradation in Germany, with data ranging from between 1990Q1 and 2019Q4. To obtain robust outcomes toward achieving our study objectives, we controlled for economic growth, primary energy consumption, and globalization. Using Fourier-based dynamic ARDL approaches, the outcomes of the Fourier ARDL long-run estimates indicate that (i) both energy productivity and globalization have negative effects on carbon emissions in Germany, and (ii) both economic growth and primary energy consumption have positive effects on carbon dioxide emissions.

These outcomes suggest that economic growth and primary energy consumption have positive effects on $CO_2$ emissions in Germany. The outcomes confirm the general consensus that increasing economic growth always requires a corresponding increase in energy demand, cited as a major source of the rising carbon emissions polluting the air, water, and biological resources and creating global warming [18]. Additionally, the increasing emissions resulting from growth, along with other factors, including climate change, have culminated in rising concentrations of particulate matter ($PM_{2.5}$) and ground-level ozone, creating negative health and environmental consequences for humanity to endure. The results validate the stated hypotheses on economic growth and primary energy consumption.

Additionally, given that available data indicate that by the end of 2021, Germany's primary energy consumption stood at 12,193 petajoules, 75% of total energy consumption was mainly from fossil sources, 6.2% consumption came from nuclear energy, and 16.1% was sourced from renewables, these outcomes imply Germany has a long way to go on its turnaround energy policy and to meet the emissions targets of the recent European Green Deal policy. However, it is refreshing to note that approximately 24.39% of Germany's energy mix came from low carbon sources as of September 2022, with renewable energy comprising 19.45%, an increase from 16.1% in 2021, while 4.93% is generated from nuclear power, falling from 6.2% since 2021, indicating Germany's commitment to its energy turnaround policy. These results support [18] who found that growth in China resulted in a catastrophic rise in $CO_2$ emissions. The key concept behind Germany's energy policies is "energy turnaround or transformation". The policy includes phasing out nuclear power by 2022 and gradually replacing fossil fuels with renewable energy.

Further, the results suggest that energy productivity and globalization have negative effects on carbon emissions in Germany. This supports a recent study published by

Odyssee–Mure, which revealed that Germany's energy efficiency data for final consumers, as measured by technical ODEX, has improved by around 1.4% per year since 2000 [58]. Further indications indicate larger gains have been achieved by households (2.2%/year), as well as 1.0%/year for the transport sector, and 1.6%/year for services, with the only industry recording a lower annual rate of energy efficiency improvements. With Germany's economic transformation described as a win–win endeavor in the light of globalization's negative effect on carbon emissions, the government is on track in terms of the implementation of its 2030 globalization roadmap, which involves (i) a response to rising protectionism, (ii) reducing the German export surplus, (iii) improving framework conditions for welfare-enhancing free trade, (iv) distributing globalization gains across the EU more widely, and (v) developing Germany as a business-friendly location.

It is regrettable that climate change, which became a major policy focus with the adoption of the Paris Agreement in 2015, only received serious consideration in 2020 [59]. This was when several countries announced their emission strategies for achieving neutrality. Policy changes have altered processes and structural changes in energy, transportation, and industrial applications. Although Germany is leading the crusade against carbon emissions in Europe, its continued use of nuclear energy is worrying, so policymakers must accelerate the process of shutting down nuclear facilities and increasing investments in renewable energy sources. The summary of the outcomes of the present study reported in Figure 6.

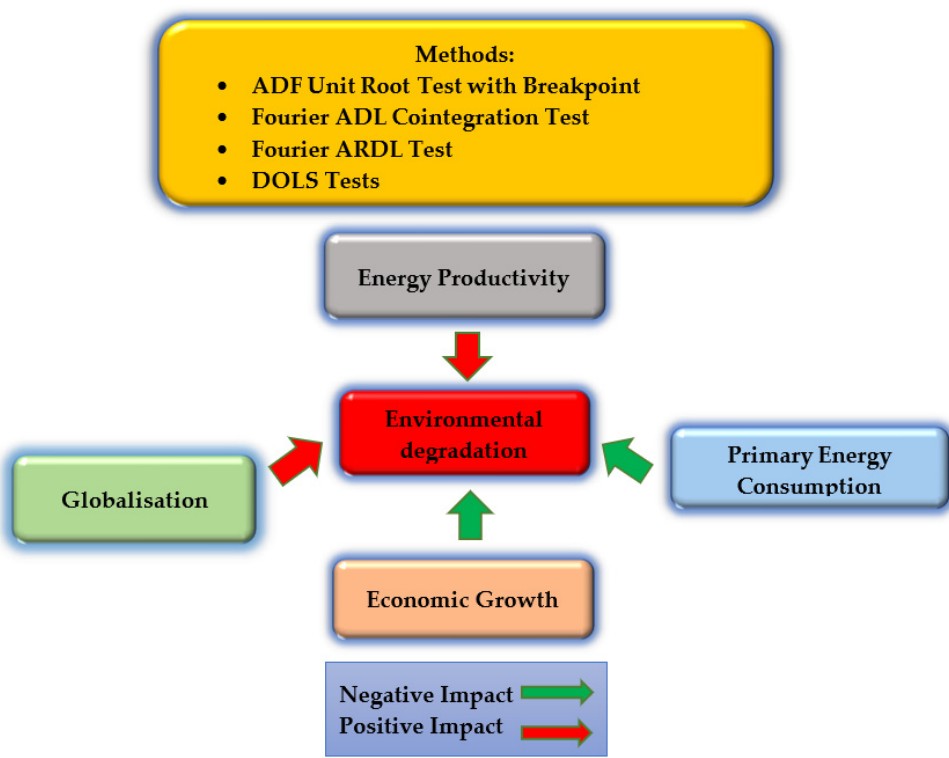

**Figure 6.** Summary of Empirical Findings with Methods.

## 6. Conclusions and Implications for Policy Action

The rise in industrialization and globalization to facilitate growth throughout the world has resulted in increased energy demand in the recent era. Industrial development requires the use of fossil fuels for production, which has adverse effects on the environment. The continuous burning of fossil fuels increases the levels of greenhouse gas emissions, leading to global warming. The resulting existential threat to humanity has generated calls for policy responses. The European Union has responded with ambitious policy directions to reduce GHG emissions. The objective of establishing a carbon-neutral economy by 2050 requires a new growth pathway, including increased intra-EU trade, reduced external trade, increased demand for local products, and low mobility.

This study has empirically investigated the effects of energy productivity on environmental degradation in Germany while controlling for economic growth, primary energy consumption, and globalization with data ranging between 1990Q1 and 2019Q4 in Germany. The outcomes of the Fourier ARDL long-run estimates indicate that (i) both energy productivity and globalization have negative effects on carbon emissions in Germany, and (ii) both economic growth and primary energy consumption have positive effects on carbon dioxide emissions. These outcomes, and the context of the EU's policy directives on a carbon-neutral economy by 2050, provide significant insights for policy action. Therefore, we make the following policy recommendations: (i) To improve its environmental record on economic globalization, Germany should implement more stringent regulatory policies to deter German car manufacturers who manipulate vehicle emission tests, as this creates inaccurate emissions data and jeopardizes human health. The admission of crime by Volkswage AG had serious social, governance, and regulatory implications for several economies. (ii) Germany must follow through with its efforts to achieve net-zero nuclear energy consumption and invest hugely in environment-friendly sourcing, i.e., renewable energy. This type of energy has proven to be clean and produces no greenhouse gas emissions. Globally, experts have warned that although establishing large-scale nuclear power generators to increase electricity could help, safety and environmental pollution mean that nuclear is not a viable alternative and needs to be phased out from the German economy. (iii) The need for a more energy-efficient economy makes it imperative for German policymakers to invest hugely into efficient energy technologies that will enable the future benefits of energy productivity to be reaped. Future researchers can consider the effects of energy productivity on environmental degradation at the regional level for comparative purposes to enable policy proposals for the EU on projects/programs that could be implemented for member economies lagging in their delivery of carbon emissions targets. This represents the limitation of the study, which only focused on a single country.

**Author Contributions:** Conceptualization, D.K.; methodology, D.K. and K.A.; software, K.A. and S.Y.G.; validation, R.A.C.; formal analysis, S.Y.G. and R.D.O.; investigation; K.A., resources, R.A.C.; data curation, K.A.; writing—original draft preparation, K.A.; writing—review and editing D.K., R.D.O. and R.A.C.; visualization, D.K.; supervision, R.A.C., G.C., R.D.O. and S.Y.G.; project administration, G.C. and D.K. All authors have read and agreed to the published version of the manuscript.

**Funding:** This paper is financed by Portuguese national funds through FCT–Fundação para a Ciência e a Tecnologia, I.P., project number UIDB/00685/2020. Also, The project is funded under the program of the Minister of Science and Higher Education titled "Regional Initiative of Excellence" in 2019–2022, project number 018/RID/2018/19, the amount of funding PLN 10 788 423,16.

**Institutional Review Board Statement:** Not applicable.

**Informed Consent Statement:** Not applicable.

**Data Availability Statement:** The variables used in this paper are collected from the database of World Bank and OECD.

**Conflicts of Interest:** The authors declare no conflict of interest.

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
