# Peer review of "Energy Productivity and Environmental Degradation in Germany: Evidence from Novel Fourier Approaches"

_sustainability, doi:10.3390/su142416911_

Round 1
Reviewer 1 Report
Comments
The paper titled ‘Energy productivity and environmental degradation in Germany; evidence from novel Fourier approaches’, investigates the effect of energy productivity on environmental degradation in Germany using data from 1990Q1 to 2019Q4. The variables used in the model are: CO2 emission, Economic growth, Globalization (KOF index), energy productivity, and primary energy consumption. So far as the write-up is concerned, it is well-written, and the flow of language is fine. I have following comments.
· Energy productivity by definition is; GDP divided by the amount of energy consumed, so it is quite intuitive that increase in energy productivity will decrease CO2 emission for the given amount of output.
· Please cite some studies concluding that there is positive relationship between energy productivity and co2 emission and give economic rationale for such relationship.
· Interestingly, for the given amount of energy used if the TFP has gone up, it would seem that energy productivity has gone up. How this question has been tackled?
· The real question is how to increase ‘energy productivity’ that unfortunately falls in the ambit of engineers not of the economists. Thus, the question that this paper investigates does not face any controversary. If the question is not controversial its investigation is less wanted.
· In the Section 2, recent studies may be incorporated. The authors are suggested to incorporate the following studies in the document cover the nonlinear relationship between economic growth and environmental quality.
1. https://doi.org/10.1016/j.esr.2022.100905
2. https://doi.org/10.1016/j.renene.2022.07.155
· Coming to the variables in the model, clarity is needed, for instance, equation 1, LGDP is economic growth, (growth rate) or natural log of GDP or GDP per capita, secondly, “the findings on LGDP invalidates the EKC hypothesis” (page 12 of 21, against row 418), how this can be concluded without introducing squared term in the model. Similarly, the KOF index has been used as a measure of globalization that is the sum of economic, social, and political globalization. What is the economics behind the relationship between KOF index and CO2 emission. The study needs to elaborate it in the text its theoretical justification. Simply citing that different studies have used this variable does not exonerate one from the onus of providing theoretical link among explanatory and dependent variable.
· Overall, the paper is fine.
Author Response
Dear Chief Editor;
We would like to thank you for considering our manuscript entitled “Energy productivity and environmental degradation in Germany; evidence from novel Fourier approaches’, investigates the effect of energy productivity on environmental degradation in Germany using data from 1990Q1 to 2019Q4. We have received invaluable comments by a reviewer, which helps us a lot to immensely improve the paper.
We hereby respond to the reviewer comments as:
PAPER:
The paper titled ‘Energy productivity and environmental degradation in Germany; evidence from novel Fourier approaches’, investigates the effect of energy productivity on environmental degradation in Germany using data from 1990Q1 to 2019Q4. The variables used in the model are: CO2 emission, Economic growth, Globalization (KOF index), energy productivity, and primary energy consumption. So far as the write-up is concerned, it is well-written, and the flow of language is fine. I have following comments.
REVIEWER COMMENTS
- # COMMENT 1
Energy productivity by definition is; GDP divided by the amount of energy consumed, so it is quite intuitive that increase in energy productivity will decrease CO2 emission for the given amount of output. Please cite some studies concluding that there is positive relationship between energy productivity and co2 emission and give economic rationale for such relationship. Interestingly, for the given amount of energy used if the TFP has gone up, it would seem that energy productivity has gone up. How this question has been tackled? The real question is how to increase ‘energy productivity’ that unfortunately falls in the ambit of engineers not of the economists. Thus, the question that this paper investigates does not face any controversary. If the question is not controversial its investigation is less wanted.
RESPONSE 1:
Thanks for your observations and comments. We have made changes in document as advised. Kindy refer to lines 107-123.
# COMMENT 2
In the Section 2, recent studies may be incorporated. The authors are suggested to incorporate the following studies in the document cover the nonlinear relationship between economic growth and environmental quality.
- https://doi.org/10.1016/j.esr.2022.100905
- https://doi.org/10.1016/j.renene.2022.07.155
RESPONSE 2:
Thanks for your observations and comments. We have made changes in literature review to reflect your suggestions. Kindly refer to line 107 – 123; & 183 & 187.
# COMMENT 3 ·
Coming to the variables in the model, clarity is needed, for instance, equation 1, LGDP is economic growth, (growth rate) or natural log of GDP or GDP per capita,
RESPONSE 3:
Thanks for your observations and comments. Equation one has been reviewed.
# COMMENT 4
secondly, “the findings on LGDP invalidates the EKC hypothesis” (page 12 of 21, against row 418), how this can be concluded without introducing squared term in the model. Similarly, the KOF index has been used as a measure of globalization that is the sum of economic, social, and political globalization. What is the economics behind the relationship between KOF index and CO2 emission. The study needs to elaborate it in the text its theoretical justification. Simply citing that different studies have used this variable does not exonerate one from the onus of providing theoretical link among explanatory and dependent variable.
RESPONSE 4:
Thanks for your observations and comments. We have made changes in literature review to reflect your suggestions. 310 - 328.
Additionally, the statement relating to “validating the EKC hypothesis” has been reviewed and taken off as it was an error.
# COMMENT 4
Overall, the paper is fine.
RESPONSE 4:
Thanks for your observations and comments. We remain very grateful for this comment.
Reviewer 2 Report
The Manuscript determines Energy productivity and environmental degradation in Germany; evidence from novel Fourier approaches. The topic is interesting and the manuscript has the potential to be published in sustainability.
However, the following major comments should be considered to improve the quality of the gray areas of the manuscript.
1. In the introduction section, importance and problems are discussed generally in good manner. But still it is limited and unclear in the perspective of Germany. It is suggested to discuss the problems in more details in the perspective of Germany.
2. The objectives, significance and contribution of the paper should be improved and written keeping Germany in mind.
3. The literature review section has explained the hypothesis very well but still it is missing with research gap. It is suggested to add a paragraph of research gap at the end of the section.
4. Section 3 should start with the heading of data and methodology.
5. Add a sub-heading of data and variables and discuss the data and description of variables. Then discuss the methodology in both theoretical and econometric way.
6. In the section of empirical findings, tables’ representation is not attractive and should follow the required format of sustainability journal.
7. Results are interpreted well, discussed with economics-based reasons and related with past studies in good manner. However, it is required to add some discussion on CUSUM and CUSUMSQ.
8. It is required to add the study's limitations and future direction at the end of the conclusion and policy recommendations with a separate heading.
9. Formatting of the overall manuscript should be on the format of sustainability journal.
10. There are some grammatical and typo errors and it is expected that the revised manuscript will be improved and error-free.
Author Response
Dear Chief Editor ;
We would like to thank you for considering our manuscript entitled “Energy productivity and environmental degradation in Germany; evidence from novel Fourier approaches’, investigates the effect of energy productivity on environmental degradation in Germany using data from 1990Q1 to 2019Q4. We have received invaluable comments by a reviewer, which helps us a lot to immensely improve the paper.
We hereby respond to the reviewer comments as:
PAPER
The Manuscript determines Energy productivity and environmental degradation in Germany; evidence from novel Fourier approaches. The topic is interesting and the manuscript has the potential to be published in sustainability.
REVIEWER COMMENTS
However, the following major comments should be considered to improve the quality of the gray areas of the manuscript.
- # COMMENT 1
- In the introduction section, importance and problems are discussed generally in good manner. But still it is limited and unclear in the perspective of Germany. It is suggested to discuss the problems in more details in the perspective of Germany
RESPONSE 1:
Thanks for your observations and comments. We have made changes in document as advised.
- # COMMENT 2
- 2.The objectives, significance and contribution of the paper should be improved and written keeping Germany in mind
RESPONSE 2:
Thanks for your observations and comments. We have made changes in document as advised.
- # COMMENT 3
- The literature review section has explained the hypothesis very well but still it is missing with research gap. It is suggested to add a paragraph of research gap at the end of the section.
Thanks for your observations and comments. We have reviewed that portion and inserted the research gap as advised
- # COMMENT 4
- Section 3 should start with the heading of data and methodology.
Thanks for your observations and comments. We have reviewed that portion as advised
- # COMMENT 5
- Add a sub-heading of data and variables and discuss the data and description of variables. Then discuss the methodology in both theoretical and econometric way.
Thanks for your observations and comments. We have reviewed that portion as advised
- # COMMENT 6
- In the section of empirical findings, tables’ representation is not attractive and should follow the required format of sustainability journal.
RESPONSE 6
Thanks for your observations and comments. The tables have been worked on
- # COMMENT 7
- Results are interpreted well, discussed with economics-based reasons, and related with past studies in good manner. However, it is required to add some discussion on CUSUM and CUSUMSQ.
Thanks for your observations and comments. We have made changes on CUSUM and CUSUMSQ results as advised. Please see line relevant lines (544 -550)
- # COMMENT 8
- It is required to add the study's limitations and future direction at the end of the conclusion and policy recommendations with a separate heading.
RESPONSE 8:
Thanks for your observations and comments. We have made changes in conclusion as advised. Kindly check.
- # COMMENT 9
- Formatting of the overall manuscript should be on the format of sustainability journal.
RESPONSE 9:
Thanks for your observations and comments. We have made reviewed the document formatting as advised
- # COMMENT 10
- There are some grammatical and typo errors and it is expected that the revised manuscript will be improved and error-free.
RESPONSE 10:
Thanks for your observations and comments. We have checked grammar errors are typos as advised
Reviewer 3 Report
Dear Editor,
I have completed the review process of the article entitled “Energy productivity and environmental degradation in Germany; evidence from novel Fourier approaches”. I found the paper promising. To further improve the quality of the paper, the authors might consider my following comments:
Even if productivity and efficiency are two different terms, as the scholars often use these two terms interchangeably, it might lead to some confusions. Please further elaborate this difference in the paper as the main emphasis of the paper is on the energy productivity.
Please critically assess the previous literature and explicitly present the contribution of the paper to the existing literature and show the novelties. I think the arguments discussed in the introduction section should be supported by some other studies.
It seems that the main contribution of the paper is mainly coming from the empirical aspects. Therefore, in the introduction section, please further discuss the empirical techniques employed and present the novelties explicitly. Hence, information provided in lines 115-117 should be improved.
The literature review section should be improved with some other previous studies that specifically focus on Germany. For example, Agnolucci P (2009) The effect of the German and British environmental taxation reforms: a simple assessment. Energy Policy 37(8):3043– 3051; AlataÅŸ, S. Towards a carbon-neutral economy: The dynamics of factor substitution in Germany. Environ Sci Pollut Res 27, 26554–26569 (2020). https://doi.org/10.1007/s11356-020-08955-2; Kemfert C, Welsch H (2000) Energy-capital-labor substitution and the economic effects of CO2 abatement: evidence for Germany. J Policy Model 22(6):641–660
What is the proxy for energy productivity?
Why does the paper specifically focus on Germany?
Improve presentation of the empirical findings. It is suggested to clearly articulate the results. Instead of discussing statistical significance, authors should discuss the economic significance of their results. Clearly discuss how the results differ from or in line with the literature and how the results are new and contribute to the body of knowledge with proper justification?
The section (3.1) provides the theoretical background of the model specification. Yet, it is not clear.
Author Response
Dear Chief Editor ;
We would like to thank you for considering our manuscript entitled “Energy productivity and environmental degradation in Germany; evidence from novel Fourier approaches’, investigates the effect of energy productivity on environmental degradation in Germany using data from 1990Q1 to 2019Q4. We have received invaluable comments by a reviewer, which helps us a lot to immensely improve the paper.
We hereby respond to the reviewer comments as:
- # COMMENT 1
Even if productivity and efficiency are two different terms, as the scholars often use these two terms interchangeably, it might lead to some confusions. Please further elaborate this difference in the paper as the main emphasis of the paper is on the energy productivity.
RESPONSE 1:
Thanks for your observations and comments. We have reviewed the document reflect energy productivity to keep the message thread as advised.
- # COMMENT 2
Please critically assess the previous literature and explicitly present the contribution of the paper to the existing literature and show the novelties. I think the arguments discussed in the introduction section should be supported by some other studies.
RESPONSE : 2
Thanks for your observations and comments. We have made necessary reviews in document to reflect contributions to literature as advised. Kindy refer to the end of the literature review (lines 275 to 286)
- # COMMENT 3
It seems that the main contribution of the paper is mainly coming from the empirical aspects. Therefore, in the introduction section, please further discuss the empirical techniques employed and present the novelties explicitly. Hence, information provided in lines 115-117 should be improved.
RESPONSE 3:
Thanks for your observations and comments. We have made the necessary reviews to include the novelty of empirical techniques employed at the introduction section as advised. Kindy refer.
- # COMMENT 4
The literature review section should be improved with some other previous studies that specifically focus on Germany. For example,
- Agnolucci P (2009) The effect of the German and British environmental taxation reforms: a simple assessment. Energy Policy 37(8):3043– 3051; AlataÅŸ, S. Towards a carbon-neutral economy: The dynamics of factor substitution in Germany. Environ Sci Pollut Res 27, 26554–26569 (2020).
- https://doi.org/10.1007/s11356-020-08955-2; Kemfert C, Welsch H (2000) Energy-capital-labor substitution and the economic effects of CO2 abatement: evidence for Germany. J Policy Model 22(6):641–660
RESPONSE 4:
Thanks for your observations and comments. We have made changes in document as advised. Kindy refer to lines 107 – 122
# COMMENT 5
What is the proxy for energy productivity?
RESPONSE 5:
Thanks for your observations and comments. We have reviewed the document as advised. Kindy refer to the methodology section
# COMMENT 6
Why does the paper specifically focus on Germany?
RESPONSE 6: Thanks for your observations and comments. The following is for your understanding and does not affect the paper.
The paper specifically focuses on Germany for several reasons:
- Germany believes energy productivity could reduce carbon emissions.
- Energy productivity is one of the three pathways outlined in their energy policy
- Germany’s “energy productivity strategy 2050" was adopted in December 2019, which prescribes new energy efficiency target for 2030
- Recently (July 2022), the country-initiated steps to design new gas auction models to encourage industrial gas savings
- In the wake of rising energy prices, the outcomes of the study could offer significant policy insights to Germany and other EU members in taking drastic policy actions amidst the use of energy as a weapon by Russian
# COMMENT 7
Improve presentation of the empirical findings. It is suggested to clearly articulate the results. Instead of discussing statistical significance, authors should discuss the economic significance of their results. Clearly discuss how the results differ from or in line with the literature and how the results are new and contribute to the body of knowledge with proper justification?
RESPONSE 7:
Thanks for your observations and comments. We have made changes in document as advised. Kindy refer to lines.
# COMMENT 8
The section (3.1) provides the theoretical background of the model specification. Yet, it is not clear.
RESPONSE 8:
Thanks for your observations and comments. We have reviewed it to give clarity as advised. Kindy refer.
Author Response
Dear Chief Editor;
We would like to thank you for considering our manuscript entitled “Energy productivity and environmental degradation in Germany; evidence from novel Fourier approaches’, investigates the effect of energy productivity on environmental degradation in Germany using data from 1990Q1 to 2019Q4. We have received invaluable comments by a reviewer, which helps us a lot to immensely improve the paper.
We hereby respond to the reviewer comments as:
REVIEWER 4
# COMMENT 1
Specific comments: 1. The relationship between energy and environmental degradation has been widely discussed in extant literature. What are their limitations? What the marginal contributions of this study?
RESPONSE 1:
Thanks for your observations and comments. We have reviewed the document to reflect your concerns. Kindly check the conclusion of the literature review (lines 279-288).
# COMMENT 2
- The abstract presents the main findings without implications; I recommend the authors say something more powerful in the abstract.
RESPONSE 2:
Thanks for your observations and comments. We have reviewed the document to include policy implications/insights as advised
# COMMENT 3
- In Introduction section, it is recommended to state clearly the research questions and why these questions or issues are important for Germany.
RESPONSE 3:
Thanks for your observations and comments. We have reviewed the document to reflect your suggestion. The hypotheses are hinted in line 63 to 69 (introduction) and further elaborated in lines 173 to 180 (literature review section). Kindly check.
# COMMENT 4
- In the second section (Review of published works of literature). After reviewing the published works of literature, you’d better summarize the possible research gaps and the contributions of this study, and why this study focuses on the proposed four hypothesizes?
RESPONSE 4:
Thanks for your observations and comments. We have reviewed the document to reflect your suggestion. Kindly refer to lines 273 to 285
# COMMENT 5
- The authors attempted to set up the ‘theoretical foundation’ of the research model (line 255 to 282). However, the relevant discussions seem quite abstract, which should be explained in a more systematic way. More detailed explanations are needed.
RESPONSE 5:
Thanks for your observations and comments. We have reviewed and improved it to reflect your concerns. Kindly refer
# COMMENT 6
- Section 3.1 presents key research indicators for exploring research questions. Readers may demand justifications on how you define them? And why they are appropriate indicators for measuring the key indicators. Are there any control variables? What are the rationales behind them? At the very least, supporting references shall be provided.
RESPONSE 6:
Thanks for your observations and comments. We have reviewed the document to reflect your suggestion
# COMMENT 7
- Some more general and insightful discussions and implications for developing benchmarks for carbon reduction from the perspective of energy productivity are expected.
RESPONSE 7:
Thanks for your observations and comments. We have reviewed the document to reflect your suggestion
# COMMENT 8
- Please update Figure 2 to improve its resolution.
RESPONSE 8:
Thanks for your observations and comments. We have reviewed the document to reflect your suggestion.
Round 2
Reviewer 3 Report
The paper can be considered for publication.
Author Response
Comment 1=The paper can be considered for publication.
Response to comment = thanks a lot for your comments in the first stage of revision
Author Response
8th December, 2022
Dear Chief Editor;
We would like to thank you for considering our manuscript entitled “Energy productivity and environmental degradation in Germany: evidence from novel Fourier approaches”. It seeks to investigate the effect of energy productivity on environmental degradation in Germany using data from 1990Q1 to 2019Q4.
We have received invaluable comments by a reviewer, which helps us a lot to immensely improve the paper. We hereby respond to the reviewer comments as:
Editorial comments:
Energy productivity and environmental degradation in Germany; evidence from novel Fourier approaches. I believe the manuscript is much improved, some minor problems still need to be addressed before publication.
- Please update the latest literature. The literature cited in the revised manuscript is too old. For example, the citation in line 265 to line 292.
Author response: Thanks for your observations and comments. Kindly be informed that the literature has been updated to include the latest literature as highlighted. The sentences were reorganized as well to straighthen the arguments. We are grateful to you for the suggestions to improve the quality of the paper.
- “Theoretically, efficient production and the productive utilization of energy promote environmental quality (Howarth, 1997). Similarly, economic globalization facilitates investments in efficient energy technologies that propel growth, which positively affects environmental pollution (Shahbaz et al., 2019).” Since it is your research theoretical foundation, please explain more and make it more compelling.
Author response: Thanks for your observations and comments. Kindly be informed that thepretical foundataion has been reviewed and updated to capture the required theoretical and empirical explanations. Please find as hightlighted green.
- Regarding the policy implication section, how the authors draw the following implication? (i) Germany should implement more stringent policies to deter German car manufacturers who manipulate vehicle emission tests, as this creates inaccurate emissions data and jeopardizes humanity.
Author response: Thanks for your observations and comments. Kindly be informed that the conclusion has been linked to the discussions on LGLO in the long-run Fourier estiates. Please this has been updated and hightlighted green for your review.
- Please update the format of all the Tables.
Author response: Thanks for your observations and comments. Kindly be informed that the tables have been updated to reflect your suggestions.
- I believe the Title should be: “Energy productivity and environmental degradation in Germany: evidence from novel Fourier approaches”. Please check edit the manuscript more carefully.
Author response: Thanks for your observations and comments. Kindly be informed that the title has been reviewed to reflect your concerns and suggestions as hightlighted green.
- Please provide version with track changes.
Author response: Thanks for your observations and comments. Kindly be informed that the reviews, corrections, and updates have been highlighted green and referenced tracked as well. Kindly check.
NB: We remain grateful to you. All your suggestiions have helped us improve the quality of the manuscript. Thank you.